# Interfacial Shear Performance of Epoxy Adhesive Joints of Prefabricated Elements Made of Ultra-High-Performance Concrete

**DOI:** 10.3390/polym14071364

**Published:** 2022-03-28

**Authors:** Kun Yu, Zhongya Zhang, Yang Zou, Jinlong Jiang, Xingqi Zeng, Liang Tang

**Affiliations:** 1State Key Laboratory of Mountain Bridge and Tunnel Engineering, School of Civil Engineering, Chongqing Jiaotong University, Chongqing 400074, China; yukun@mails.cqjtu.edu.cn (K.Y.); zhangzhongya@cqjtu.edu.cn (Z.Z.); jinlongjiang@mails.cqjtu.edu.cn (J.J.); 980201100098@cqjtu.edu.cn (X.Z.); tangliang@cqjtu.edu.cn (L.T.); 2Department of Civil Engineering, School of Civil and Transportation Engineering, Shenzhen University, Shenzhen 518060, China

**Keywords:** shear performance, experimental study, adhesive joint, ultra-high-performance concrete (UHPC), passive constraint

## Abstract

Application of ultra-high-performance concrete (UHPC) in joints can improve the impact resistance, crack resistance, and durability of structures. In this paper, the direct shear performance of ultra-high-performance concrete (UHPC) adhesive joints was experimentally studied. Twenty-four direct shear loading tests of UHPC adhesive joints were carried out considering different interface types and constraint states. The failure modes and load-slip curves of different interfaces were studied. Results indicated that passive confinement could enhance the strength and ductility of the interface; the average ultimate bearing capacity of the smooth, rough, grooved, and keyway specimens with passive restraint were, respectively, increased by 11.92%, 8.91%, 11.93%, and 17.766% compared with the unrestrained ones. The passive constraint force changes with the loading and finally tends to be stable. The epoxy adhesive has high reliability as a coating for the UHPC interface. The adhesive layer is not cracked before the failure of the specimen, which is also different from the common failure mode of adhesive joints. Failure of all specimens occurred in the UHPC layer, and the convex part of the groove interface shows the UHPC matrix peeling failure; the keyway interface is the shear damage of the key-tooth root, and the rest of the keyway showed UHPC surface peeling failure. According to the failure mode, the shear capacity of UHPC keyway adhesive joints under passive restraint is mainly provided by the shear resistance of key teeth, the friction force of the joint surface, and the bonding force of the UHPC surface. The friction coefficient was determined based on the test results, and the high-precision fitting formula between the shear strength of the UHPC surface and the passive constraint force was established. According to the Mohr stress circle theory, the proposed formula for direct shear strength of UHPC bonded joints under passive constraint was established. The average ratio of the proposed UHPC adhesive joint calculation formula to the test results was 0.99, and the standard deviation was 0.027.

## 1. Introduction

Ultra-high-performance concrete (UHPC) is a new type of cement-based composite [1], which is usually composed of cement, silica fume, quartz sand, fiber, superplasticizer, and other components [2]. UHPC has the characteristics of high strength, high toughness, high ductility [3], crack resistance [4], impact resistance [5], and durability [6]. Components built with this material have lightweight, strong spanning ability [7,8] and have become a new type of building material with application prospects [9]. However, UHPC has a very low water–cement ratio. [10,11]. The very low water–cement ratio of UHPC is also considered the main reason the shrinkage cannot be ignored, especially in the case of poor curing conditions [12]. Due to the above characteristics of UHPC, in engineering applications, the extensive use of UHPC in cast-in-place mode is limited, and the use of UHPC prefabricated components can well avoid the above problems [13,14], with good curing conditions in prefabricated plants.

Studies have shown that using UHPC as a prefabricated component can enhance structural fatigue performance, service performance, ductility, and stiffness performance in prefabricated structures and reduce the structural weight [15]. Further, according to French R et al. [16], the addition of fibers significantly increased the peak shear stress and the corresponding shear slip of concrete. The prefabricated bridge deck joints are divided into wet joints and dry joints. Hussein et al. [17] studied the mechanical properties of UHPC as a structural wet joint material. They believed that UHPC wet joints had excellent mechanical properties regardless of whether the shear reinforcement was configured. Jang et al. [18] studied the direct shear performance of Z-shaped UHPC integrally poured flat (wet) joint specimens. The results showed that the interfacial shear strength of the specimens chiseled with high-pressure water jets could reach 32.2% of the overall specimens. However, in some extreme environments where the prefabricated bridge deck assembly joints cannot meet the maintenance conditions, the shrinkage of UHPC itself will lead to poor connection performance of the wet joints. Therefore, adhesive joints are more suitable in extreme environments than in wet joints, and their application fields are also relatively wide. Allan Manalo and other scholars carried out the axial compression test of modular composite walls and the bonding behavior test of the composite sandwich plate and epoxy adhesive and proposed a reliable theoretical equation for epoxy joints. The epoxy joint has been considered for use in wall systems [19] and sandwich panels [20] and other fields. The performance of adhesive joints is also excellent, as C. H. Lee et al. [21] conducted tests on UHPC direct shear specimens with different joint types and found that the bearing capacity of adhesive joints is higher than that of dry and wet joints. However, adhesive joints show greater brittleness than dry joints [22,23]. In addition, studies have shown that the optimal design of structural joint details has obvious effects on the mechanical characteristics of the structure in the construction stage, the normal operation stage, the maintenance and repair stage, and the bearing capacity limit state. Therefore, it is particularly important to study the shear performance of segmental joints.

At present, there are many research results on dry joints. YL Voo et al. [24] conducted experiments on dry joint direct shear specimens with different key teeth and normal stress levels. They established a direct shear strength calculation formula suitable for UHPC dry joint calculation. The results of cemented joints are still in constant accumulation. Buyukozturk O et al. [25] studied flat and key groove joints, no epoxy (dry) and prestress level, epoxy thickness, and other parameters. The results show that the strength of epoxy joints is always higher than that of dry joints. There was no direct relationship between bonding strength and adhesive layer thickness. However, in the subsequent studies, Gopal B A et al. [26] conducted parametric studies on the influence of the number of shear keys, lateral stress, node type, and other factors on the shear capacity of joints. The results showed that the bearing capacity of UHPFRC bond joints increased with the increase of the horizontal pressure (confining pressure) applied to the joints, the number of shear keys, and the thickness of epoxy resin used on the joints. Yuan et al. [27] found that the failure mode of adhesive joints may occur as concrete failure along the edge of joints, epoxy resin failure, and bond failure between concrete and epoxy resin. While the failure of dry joints always occurs, brittle fractures of key teeth fall off along the joints. Whether dry or adhesive joints, the shear capacity of joints increases with lateral pressure. Therefore, the lateral pressure positively impacts the bridge deck joint. Zhou et al. [22] found that the reduction effect of key teeth should be considered in the stress of multi-key-tooth dry joints, and the shear capacity of adhesive joints has no direct relationship with the number of key teeth. Then, Chen et al. [28] conducted a full-scale shear test considering the joint form and the number of shear keys. The study found that the shear bearing capacity and plastic deformation capacity of single-bond epoxy joints improved compared with plain epoxy joints and integral joints. The number of key teeth positively affects shear capacity and plastic deformation capacity. Li et al. [29] found that the failure of the dry joint of the key-tooth interface is the root shear of the key tooth, and the failure of the epoxy joint is the concrete failure near the joint. C. H. Lee et al. [21] and Y. J. Kim et al. [30], in the direct shear test of UHPC adhesive joints, found that the depth of key teeth would improve the shear capacity of joints. Gopal et al. [31], through the direct shear test of UHPC dry joint and adhesive joint, found that increasing the number of key teeth will reduce the influence of epoxy resin and normal stress level on the shear strength of the joint. The calculation formula of the bearing capacity of the glued joint considering the influence of the positive bond strength of the glue layer was given.

In summary, the research results of the shear performance of ordinary concrete bridge deck joints and UHPC dry joints are more remarkable. The existing research on adhesive joints mainly focuses on the interface of UHPC-NC and UHPC-UHPC bond-slot considering the shear performance under unconstrained and applied active stress conditions. However, in many practical projects (such as bridge head position joints and modular wall system joints, etc.), the interfacial shear will be subjected to the passive constraint from surrounding structures due to the “shear effect”. At present, the test sample data of adhesive joint performance under passive constraints is not enough; the shear failure mechanism, the contribution of the adhesive layer, and the failure mode of UHPC adhesive joints under passive constraints are unclear.

This study is aimed at the shear performance of UHPC adhesive joints to clarify the shear transfer mechanism of the joint under different restraint states and the contribution of the adhesive layer to the joint shear strength. The two main influencing factors of joint type and passive restraint are used as test parameters in this paper. The direct shear test of 24 UHPC adhesive joint specimens was completed. The direct shear failure characteristics, initial crack strength, peak strength, interface stiffness, and shear bearing capacity of UHPC specimens under different interfaces and passive constraints were studied. The shear bearing capacity calculation formula is proposed based on the Mohr stress circle principle. This formula can better predict the ultimate bearing capacity of UHPC adhesive joints under passive constraints. This study provides a calculation basis and reference for designers and researchers in the field of UHPC precast adhesive joints.

## 2. Test

### 2.1. Specimen Design

To study the interface performance and failure mode of the epoxy bonding interface of prefabricated UHPC adhesive joints under passive constraints, the author designed the natural pouring (smooth) interface, rough grinding interface, groove interface, and keyway interface according to several commonly used interface types. Considering that the interface shearing will be constrained by the surrounding structure due to the “shear expansion effect” in actual engineering (such as composite beam concrete bridge deck and modular wall system etc.), the methods of unconstrained loading and passive constrained loading were, respectively, set. Eight groups of shear experiments were carried out according to different loading methods and interface types, and each group had three specimens for a total of 24 specimens.

The specimen parameters are shown in Table 1, and the details of specimens are shown in Figure 1. To avoid loading eccentricity, direct shear specimens (DS-S, DS-R, DS-G, DS-K, PC-DS-S, PC-DS-R, PC-DS-G, PC-DS-K) were formed by splicing two L-shaped specimens, which can control the loading point to be located in the plane where the interface is located. The area of the interface of each specimen is 150 × 150 mm^2^, and the thickness of the epoxy layer is 2 mm. DS represents direct shear, PC represents passive restraint, and S, R, G, and K represent smooth interface, rough interface, groove interface, and keyway interface, respectively. The reinforcement diagram of the specimen is shown in Figure 2.

The production process of the test piece is as follows: UHPC was cast in place and then cured at room temperature for 48 h; after curing, the mold was removed and cured by steam at 95 °C for 48 h. The interface of the specimen was bonded by epoxy adhesive, and the specific treatment method was as follows: The interface of the natural pouring specimen was not specially treated. After the strength was formed, the interface part was cleaned with acetone. For a rough interface, we used an alloy hammer to smash the UHPC on the interface to expose some steel fibers and then used acetone to clean the surface. For the groove interface, we stuck a wooden strip with a thickness of 2 mm and a width of 10 mm on the formwork for pouring and cleaned the groove surface after the formwork was removed. The concave side was first poured for the keyway interface to ensure the specimen’s accuracy. After the strength of the UHPC was formed, the template of the concave interface was removed, a plastic film was laid on the surface, and then, the convex side template was placed on it for secondary pouring at room temperature. After curing for 48 h, steam curing was carried out. After all the cleaning was completed, the quantitative epoxy adhesive was wiped evenly from the center of the two interfaces to the surrounding areas. The thickness of the epoxy layer was distributed as thick in the middle and thin on both sides. Before the strength of the epoxy adhesive was formed, the UHPC interface was closely attached in time and squeezed with appropriate force until the colloid overflowed a little from the edge. The thickness of the interface was controlled at 2 mm. Finally, the specimens were cured at room temperature for seven days until the interface developed strength.

### 2.2. Material Properties

The specific mixing ratio of UHPC materials (Hunan Solid Engineering New Materials Co., Inc., Jinzhou New District, Changsha city, Hunan Province, China) used in this test is given in Table 2. Among them, the steel fibers incorporated in UHPC are straight, 8 mm long, 0.12 mm in diameter, and 2% by volume. Control cubes of 100 mm × 100 mm × 100 mm and dog bone specimens of 30 mm × 30 mm sections were prepared in the same batch. In this study, the actual compressive strength, tensile strength, and elastic modulus of UHP concrete were obtained before the test, as shown in Figure 3. According to the test standards and recommendations of the concrete engineering series, the compression and tensile test analysis of the cube and dog bone specimens were carried out, respectively. The mechanical properties of UHPC materials and CBSR-A/B epoxy resins (Carbon Technology Group Co Inc., Tianjin, China) are given in Table 3. The bolts are M16 bolts of grade 8.8, the elastic modulus is 210 GPa, and the Poisson’s ratio is 0.31.

### 2.3. Loading Scheme

The direct shear test was carried out by an electronic universal testing machine (Ruixuan Electronic Technology ( Shanghai ) Co Inc., Shanghai, China) with a range of 3000 KN. Before loading, the specimen was strictly aligned by a laser level to prevent eccentric loading. Two dial gauges (Donghua Testing Technology Co Inc., Jingjiang City, Taizhou City, Jiangsu Province, China) were symmetrically arranged in the middle on both sides of the interface of each specimen, the measurement accuracy is one-thousandth of a millimeter, and the interface slippage is the average value of the data in the two tables. After the unconstrained specimen was ready, it was directly loaded according to the loading regime, as shown in Figure 4A. For passive constrained specimens, to simulate the real stress state of the interface under passive constraint, a Q420 steel plate of 230 × 140 × 20 mm^3^ and four M16 fine thread bolts of 8.8 grade were used to constrain the specimens. Strain gauges were used to measure the microstrain of four bolts, as shown in Figure 4B. All specimens were preloaded to 50 KN before formal loading and pre-loaded according to each stage of 5 KN. The displacement loading control was used when legal loading, and the loading speed was 0.05 mm/min. The load, slip, and bolt strain were counted simultaneously during the loading process, and the test phenomenon was recorded until the specimen was loaded and failed.

## 3. Analysis of Test Results

### 3.1. Interface Failure Pattern

Loading phenomenon of specimen: faint crackling sounds appeared intermittently at the interface in the initial stage of unconstrained specimen loading, but no interfacial slippage and UHPC surface cracking were observed. When the specimen was close to reaching the ultimate bearing capacity, local cracking appeared at the interface. Then, the interface suddenly cracked after reaching the ultimate bearing capacity, and the bearing capacity of the specimen was instantly lost.

When the passive constraint specimen was loaded, the interface emitted a weak sound; near the ultimate bearing capacity, local cracks appeared at the interface; when the ultimate bearing capacity was reached, the interface suddenly cracked, and the bearing capacity did not disappear rapidly. Instead, with the increase of displacement, the shear bearing capacity was provided by the interface friction and UHPC residual bearing capacity.

In the direct shear test, there are several failure modes of the UHPC epoxy bonding interface: (I) stripping damage of UHPC surface, (II) matrix failure of UHPC, (III) local shear failure of epoxy layer, and (IV) fracture failure of the epoxy layer itself.

Unconstrained DS-S specimen had a slight noise when loading and a large noise when failure occurred, but no cracks were observed before failure occurred. The typical failure mode of this kind of specimen is shown in Figure 5A. The failure was manifested as the peeling failure of (I) the UHPC surface layer, (II) the failure of the UHPC surface matrix, and the fracture of a small amount of (IV) epoxy adhesive layer. The loading and failure phenomena of confined PC-DS-S specimens are similar to those of unconstrained specimens, and slight cracks were observed before failure. The typical failure modes of such specimens are shown in Figure 5B. On one side, the epoxy layer was relatively intact. On the other side, much (I) UHPC surface delamination damage and a small amount of (II) UHPC surface matrix damage occurred. Obvious UHPC damage was observed on the bonding interface of joints after failure.

The unconstrained DS-R specimens had slight noises during loading and failure, and no obvious cracks were observed before the collapse. The typical failure mode of this type of specimen is shown in Figure 5C. The failure shape is that one side of the epoxy layer was intact, and the other side appeared to show (II) UHPC matrix failure. The loading and failure phenomenon of the constrained PC-DS-S specimen is similar to that of the unconstrained specimen, and the typical failure mode of this kind of specimen is shown in Figure 5D. The damage manifested as an intact epoxy layer on one side, a very small amount of (I) UHPC surface peeling damage on the other side, and (II) UHPC surface matrix damage in a large area. After damage, obvious UHPC damage was observed on the bond and detachment surface of the seam. The failure types of the rough interface specimens were the same, and a large number of UHPC peeling occurred at the interface. The reason is that the rough treatment of the UHPC interface leads to slight damage on the surface. The interface cracks extended along the damaged surface and finally completely separated.

When loaded, the unrestrained DS-G specimens had slight noises, and violent vibrations and loud noises appeared during failure. No obvious cracks are observed before failure. The typical failure mode of this type of specimen is shown in Figure 5E. The failure shape is that one side of the epoxy was relatively intact, and the other side had (I) UHPC surface peeling damage and (II) UHPC matrix damage; the UHPC groove’s protruding part was sheared. The loading and failure phenomena of constrained PC-DS-G specimens are similar to those of unconstrained specimens, but slight cracks were observed before failure. The typical failure modes of such specimens are shown in Figure 5F. The failure was manifested, as one side of the epoxy was relatively intact. The other side had a very small amount of (I) surface peeling failure of UHPC, (II) matrix failure of UHPC—a shear failure of the prominent part of the UHPC groove—and a very small amount of (III) local shear failure of the epoxy layer. After damage, obvious UHPC damage was observed on the bond and detachment surface of the seam. The difference between the failure types of groove-constrained and unconstrained specimens is that the unconstrained specimens have large UHPC matrix peeling failures, and a small amount of epoxy is sheared. The bearing capacity of the specimen after failure was mainly provided by the residual force and frictional force of UHPC.

The unconstrained DS-K specimen had a slight noise when loaded, and there was a violent vibration and a large, muffled sound when it was damaged. Before the failure, a tiny crack was observed at the top and bottom of the interface and gradually expanded to the keyway. The typical failure form of this type of specimen is shown in Figure 5G, the crack is shown in Figure 5I, and all the concave sides of the keyway are not damaged. The failure shape is that one side of the epoxy was relatively intact, and the other side had (I) UHPC surface peeling failure, (II) UHPC matrix failure-bond teeth, and UHPC failure within a certain range of its roots; and (III) the epoxy layer itself fractured and failed. The loading and failure phenomenon of the constrained PC-DS-K specimen is similar to that of the unconstrained specimen, and the crack development was basically the same. The typical failure mode of this type of specimen is shown in Figure 5H. The damage manifested as a relatively intact epoxy layer on one side, (I) UHPC surface peeling damage on the other side, (II) UHPC matrix damage as key was sheared, and (III) the epoxy layer itself was fractured and damaged. After the damage, obvious UHPC damage was observed on the bond and detachment surface of the seam.

According to the failure mode, it can be seen that: (1) For all specimens, most of the damage was concentrated in the UHPC layer, mainly the surface stripping damage of UHPC and the peeling failure of the UHPC matrix. It indicated that the epoxy adhesive as the coating of the UHPC interface has high reliability. For unconstrained specimens, many UHPC matrix debonding failure occurs at the interface. This phenomenon is because when the interface is about to reach the ultimate bearing capacity, there will be fine cracks on the surface of UHPC. When the interface continues to be loaded, due to the strong bonding force between UHPC and epoxy, the cracks will develop along the principal stress direction inside UHPC. When the interface reaches the ultimate bearing capacity, the interface suddenly cracks due to the release of internal energy, resulting in many UHPC matrix peeling failures. For passively restrained specimens, the failure mode is mainly concentrated in the peeling failure of the UHPC surface. This phenomenon is because when the interface is about to reach the ultimate bearing capacity, there will be fine cracks on the surface of UHPC. When the interface continues to load, due to passive constraints, the transverse development of cracks is limited, and most of them only develop along the surface of UHPC. When reaching the ultimate bearing capacity, the interface will not suddenly crack due to the role of passive constraints but gradually expand from the edge to the middle. This is also why the ultimate bearing capacity of passive confined specimens is higher than that of unconstrained specimens.

(2) According to the two main failure modes, the contribution of the adhesive layer to the interface can be expressed as (I) type of failure mode, the interface is subjected to tangential slip, and the bond strength of the adhesive layer provides resistance to the load. When the UHPC surface reaches the maximum bonding strength, the UHPC surface, and the epoxy adhesive layer begin to be damaged, the shear contribution of the bonding force to the interface decreases, and the frictional force in the damaged area begins to provide shear stress. Therefore, when the interface is damaged, the interfacial shear resistance should result from the combined effect of the effective bonding strength of the UHPC layer and the frictional shear resistance; (II) type of failure mode can be attributed to the UHPC failure when the UHPC reaches the maximum shear strength, the epoxy adhesive. When the layer does not obtain the maximum bond strength, the UHPC is damaged, and the shear resistance should be the result of the combined effect of UHPC residual strength and frictional shear resistance.

### 3.2. Load-Slip Curve

The load-slip curves of the specimen in the whole loading process are plotted in Figure 6. It can be seen that before cracking, the overall shear of the specimen, the load-slip curve, is roughly linearly increased. When the specimen reaches the cracking load $V_{c}$, subtle cracks appear. A short “platform” is observed on the load-slip curves of some unconstrained specimens, and the sliding “platform” of passive, constrained specimens is relatively long. As the external load increases, the ultimate failure load $V_{b}$ is quickly reached. The crack develops rapidly until the interface is suddenly destroyed, and then, the bearing capacity of the specimen drops sharply. There is an obvious downward mutation process. However, the load-displacement curves of all specimens after failure were not obvious, mainly because the sudden release of energy in the interface led to severe vibration of the specimen. The dial indicator could not capture the slip data at this stage. Unconstrained specimens enter failure mode after interface cracking. The interface is completely detached; when the constrained specimen continues to load, the load can increase, but it has not reached the $V_{b}$ value. This process is as follows: after the load reaches the $V_{b}$ value, the crack has fully developed, but due to the existence of passive constraints, the interface is not completely separated. The load value decreases as the partially cracked UHPC exits the shearing work. When the specimen is under the lateral passive constraint, the shear bearing capacity can continue to increase due to the dislocation of the failure surface. When the specimen is dislocated to a certain extent, the frictional force provided by the interface has reached the limit, and the load cannot be further increased.

The loading process of unconstrained direct shear specimens can be divided into three stages: the first stage is still the linear elastic deformation stage before failure; the second stage is the crack development stage after specimen cracking. The surface UHPC cracking causes a decrease in specimen stiffness and an increase in relative slip. The third stage is the failure stage of the specimen. The first three stages of the loading process of the constrained direct shear specimen are the same as those of the unconstrained direct shear specimen, and the fourth stage is the staggered slip stage.

For smooth and rough interfaces, as shown in Figure 6A,B, from the load-slip curve, all specimens have no obvious yield stage after cracking, and the load value decreases immediately after cracking, which is attributed to brittle failure. Due to the existence of constraint force, the passive specimen still has a certain residual bearing capacity after failure.

For the groove interface, as shown in Figure 6C, from the load-slip curve, all specimens will enter the short slip “platform” after cracking, and the load value of some specimens will increase after cracking. There will be a certain residual bearing capacity at the loading interface after failure, and the residual bearing capacity has no obvious law compared with other passive constraint specimens. The main reason is that the epoxy layer is relatively intact after the failure of the groove interface. In contrast, the UHPC interface is rather rough after failure, providing a large friction force for the interface.

Figure 6D shows that all specimens will enter a short-term slip “platform” after cracking from the load-slip curve for the keyway interface. After expansion, finally, the key teeth are sheared and destroyed. After the failure of the unconstrained specimen, the interface was not completely disengaged, and there was still some residual bearing capacity.

### 3.3. Interfacial Bonding Strength

The interfacial bond strength is one of the key indicators to evaluate the bonding performance of the interface. Because the randomness of the distribution of steel fibers in UHPC leads to differences in the shear strength of the interface, and the bond strength of each position on the bonding interface is not the same, the average bond shear strength of the interface was selected for analysis. According to Formula (1), the average bond shear strength of the interface was calculated:*τ* = *F*/A(1)

In the formula, τ is the interface average bond shear strength (MPa); f is the cracking load, ultimate load, and interface residual load (kN) of the specimen, where residual load takes the relatively stable load after the interface failure; A is the bonding area (mm^2^) of UHPC-UHPC direct shear specimens. The calculated interface shear cracking strength and standard deviation of ultimate strength are shown in Figure 7A,B, and the residual interface strength is shown in Figure 7C. The detailed results of specimen loading are given in Table 4.

According to the load-slip curve and the interface strength, the average ultimate shear strengths of the smooth, rough, grooved, and keyway specimens with passive restraint are 11.92%, 8.91%, 11.93%, and 17.76% higher than those of the unrestrained specimens, respectively. The specimen’s cracking strength and ultimate strength are equal under the condition of unconstrained loading, and the strength is more consistent. Under passive restraint loading, the average ultimate strength of smooth, rough, grooved, and keyway specimens was 3.43%, 13.75%, 23.28%, and 10.17% higher than the average cracking strength, respectively. There is a large difference between the ultimate strength and the cracking strength of the groove interface with passive restraint. The reason is that the failure state of the groove interface is complex, and the three specimens have no obvious regularity. The keyway interface has only one more keyway compared with the smooth interface. Still, its average ultimate shear strength is 1.14 times that of the smooth interface without restraint and 1.16 times that of the passive restraint loading condition.

Therefore, passive restraint has an enhancement effect on the strength and ductility of the interface, and the enhancement effect of passive restraint on the keyway interface is better than the others.

### 3.4. Passive Constraint Force-Slip Curve

The passive constraint force is calculated according to the bolt strain, and the bolt strain and load slip are synchronously collected. According to the passive constraint force–slip curve in Figure 8, it can be seen that the specimen will suffer a small passive constraint force before the interface is about to crack. After reaching the cracking load, the passive constraint force will increases with the increase of the cracking deformation of the specimen. When the interface is broken, the passive binding force will suddenly rise. This is because the interface is bound by the “shear expansion effect” under the constraint of the steel plate. The passive constraint limits the development of interface cracks and strengthens the interface. The standard deviation of passive constraint force when each interface is damaged is shown in Figure 9. When the interface is damaged, the steel plate limits the lateral movement of the specimen. The passive constraint force will increase linearly with the interface sliding further loading. The ratio between the passive constraint force and the bearing capacity after failure shows a decreasing trend, mainly because the interface is not immediately detached. There is still a certain residual force under the action of passive constraint. Continuing to the load, the proportion of residual bearing capacity of the interface gradually decreases, and the proportion of friction force gradually increases. Finally, the passive binding force and the bearing capacity will tend to a relatively stable ratio after failure. In addition, the slope of the passive constraint force and slip curve and the size of the passive constraint force are different, but the smooth interface and the keyway interface are more stable, and the groove interface is more discrete. This is because the interface failure patterns of the same specimens are not the same, and their roughness is also different.

## 4. Analysis of Test Results

### 4.1. Basic Assumptions

Since the damage degree of the interface is difficult to define, simplified treatment of interface behavior according to experimental phenomena is necessary. The surface spalling failure of UHPC in Class (I) and the matrix failure of UHPC in Class (II) are essentially UHPC failure; the proportion of local shear failure of class (III) epoxy layer is small and can be ignored. The fracture failure of the epoxy layer in Category (IV) is caused by the shear failure surfaces on both sides that are not on the same side. This kind of failure does not affect the calculation of shear bearing capacity, which can be ignored. Based on the above failure state, the following basic assumptions are introduced: (1) according to the interface failure pattern, interface strength is controlled by UHPC tensile strength; (2) the passive constraint interface is compressed after cracking, and the friction contact area on the flat part is not separated; (3) uncracked UHPC surface contribution is provided by the combined effect of tensile strength reduction and friction contribution of UHPC, and cracking the UHPC layer only provides friction contribution; and (4) it is assumed that the root shear strength of the key-tooth specimen is controlled by UHPC tensile strength and passive constraint normal stress.

### 4.2. Calculation Method

According to the current research results [31] and the results of this test, it can be considered that the shear strength V of the UHPC keyway adhesive joint is mainly provided by the key-tooth shear resistance $V_{K}$, the joint surface friction $V_{sm}$, and the UHPC surface adhesion $V_{e}$. The expression is shown in Formula (2):(2)$$V=V_{K}+V_{sm}+V_{e}$$

Shear contribution of key teeth is represented by $V_{K}$. Most of the existing research is based on the maximum principal tensile stress theory to carry out the stress analysis of the key-tooth joint and take part in the microelement analysis of the key tooth. The Mohr circle principle is used in this study, and the stress diagram is shown in Figure 10. Then, according to the calculation formula of the principal tensile stress $\sigma_{11}$ of the Mohr stress circle (3):(3)$$\sigma_{11}=\frac{\sigma_{x}+\sigma_{y}}{2}+\sqrt{{\left(\frac{\sigma_{x}+\sigma_{y}}{2}\right)}^{2}+\tau_{xy}{}^{2}}$$
where $\sigma_{x}\;$=$\;\sigma_{n}$, and $\sigma_{y}=0$. For limit state design, $\sigma_{11}=f_{t}$, where ft is the uniaxial tensile strength of the UHPC material. Therefore, the shear strength ($\tau_{xy}$) can be calculated as follows (4):(4)$$\tau_{xy}=\sqrt{{\left(f_{t}+\frac{\sigma_{n}}{2}\right)}^{2}-{\left(\frac{\sigma_{n}}{2}\right)}^{2}}$$

Then, the shear bearing capacity of the key teeth can be written as Formula (5):(5)$$V_{k}=A_{k}\tau_{xy}$$

$A_{k}$ represents the area of the root of the key tooth.

Interface friction force is represented by $V_{sm}$; the friction force is related to the normal stress, and its formula can be written as (6):(6)$$V_{sm}=\mu A_{sm}\sigma_{n}$$

$A_{sm}$ is the area of the smooth part of the interface. μ is the static friction coefficient between the concrete and the concrete surface. Its expression is fitted according to the data law of the descending segment after the failure of the keyway interface, and the presentation of μ is obtained as (7):(7)$$\mu =1.12417-0.1214\sigma_{n}$$
to obtain the Formula (8):(8)$$V_{sm}=\left(1.12417-0.1214\sigma_{n}\right)A_{sm}\sigma_{n}$$

The adhesion force of the UHPC surface layer is $V_{e}$. According to the failure mode of the smooth interface specimen, since the fiber distribution of the UHPC surface layer is different from that of the UHPC matrix, the tensile strength of the surface layer is quite different from that of the matrix. Taking the average shear strength $\tau_{b}=8.158$ MPa of the smooth interface without normal stress, the UHPC strength reduction value is φ=0.5827. According to the contribution of interfacial friction $V_{sm}$, the UHPC surface bond strength of three passive constrained smooth interface specimens is obtained. Considering the contribution of normal stress level to the shear strength of UHPC, the fitting curve shown in Figure 11 is obtained, and the formula is obtained (9):(9)$$\tau =\varphi f_{t}\left(1+0.09143\sigma_{n}-0.02763\sigma_{n}{}^{2}\right)$$

Thus, the surface bonding force expression of UHPC is (10):(10)$$V_{e}=\varphi f_{t}\left(1+0.09143\sigma_{n}-0.02763\sigma_{n}{}^{2}\right)A_{sm}$$

The formula for calculating the bearing capacity of smooth interface and keyway interface is as follows (11):(11)$$V=A_{k}\sqrt{{\left(f_{t}+\frac{\sigma_{n}}{2}\right)}^{2}-{\left(\frac{\sigma_{n}}{2}\right)}^{2}}+\mu A_{sm}\sigma_{n}+\varphi f_{t}\left(1+0.09143\sigma_{n}-0.02763\sigma_{n}{}^{2}\right)A_{sm}$$

Due to the interface’s initial damage caused by the interface’s roughness treatment, the interface bearing capacity of the rough interface specimen is greatly different from that of the smooth interface. In addition, the failure mode of the groove interface is complex, and the passive constraint force has no obvious regularity. Therefore, the author did not put forward the calculation formula of the bearing capacity of the rough interface and groove interface.

### 4.3. Comparison of Suggested Formulas with Experimental Results

According to the interface shear contribution of the above three parts, the experimental results and the calculated values of the proposed calculation model were compared and studied to evaluate the proposed computational model’s reliability. The results are shown in Table 5. It can be seen from the table that the suggested formula can better predict the shear strength of UHPC smooth joints and key-tooth joints. The ratio of the calculated value to the experimental value is 0.996, and the standard deviation is 0.027. The shear capacity of the smooth interface is mainly provided by UHPC surface adhesion and friction. The shear bearing capacity of the keyway specimen is primarily provided by the surface bonding force of UHPC, the shear resistance of the tooth, and the interfacial friction. The shear contribution of specific components of joints is shown in Figure 12.

## 5. Conclusions

Aiming at the mechanical properties of UHPC adhesive joints, this study designed and completed the direct shear test of 24 UHPC adhesive joint specimens with the interface type and passive constraint as the research parameters. The failure mode and shear performance of adhesive joints were obtained. A formula for calculating the shear strength of UHPC adhesive joints under passive constraint is proposed by the Mohr stress circle method, and the following conclusions were obtained:The failure mode of all specimens is UHPC layer failure, indicating that epoxy adhesive as coating of UHPC interface has high reliability. The smooth interface is the delamination failure of surface UHPC; the groove interface is UHPC matrix failure; the bond groove interface is the shear failure at the root of the bond tooth, and the plane part is shown as the surface stripping of UHPC.The loading process of unconstrained direct shear specimens can be divided into three stages: linear elastic deformation stage, crack development stage after cracking, and specimen failure stage; the loading process of passive constrained direct shear specimen can be divided into four stages: linear elastic deformation stage, crack development stage after cracking, specimen failure stage, and dislocation slip stage.Passive confinement enhances the strength and ductility of the interface. The average ultimate bearing capacity of smooth, rough, grooved, and keyway specimens with passive constraint is 11.92%, 8.91%, 11.93%, and 17.766% higher than specimens without constraint. The average ultimate shear strength of the keyway interface is 1.14 times that of the smooth interface without restraint and 1.16 times that of passive restraint loading. Therefore, the keyway interface is more recommended in these four types of interfaces.The passive restraint force varies with loading, rising abruptly at cracking and then increasing roughly linearly. After that, the passive binding force and the bearing capacity after failure will tend to a relatively stable ratio.The friction coefficient was determined based on the test results, and the fitting formula between the shear strength of the UHPC surface and the passive constraint force was established. A procedure for calculating the direct shear strength of UHPC glued joints is proposed based on the Mohr stress circle theory. The ratio of the computed value of the proposed formula to the experimental value is 0.996, and the standard deviation is 0.027; it indicates that the force model proposed in this paper can be used to estimate the shear strength of UHPC smooth joints and keyway joints under passive constraints.The bearing capacity of the keyway interface under unconstrained conditions is provided by the surface bonding force of UHPC and the shear strength of key teeth, and the contribution of key teeth is 30%. Under passive constraint, the adhesive force of the UHPC surface accounted for 66%, the contribution of key teeth accounted for 27.6%, and the rest was contributed by friction.

## Figures and Tables

**Figure 1 polymers-14-01364-f001:**
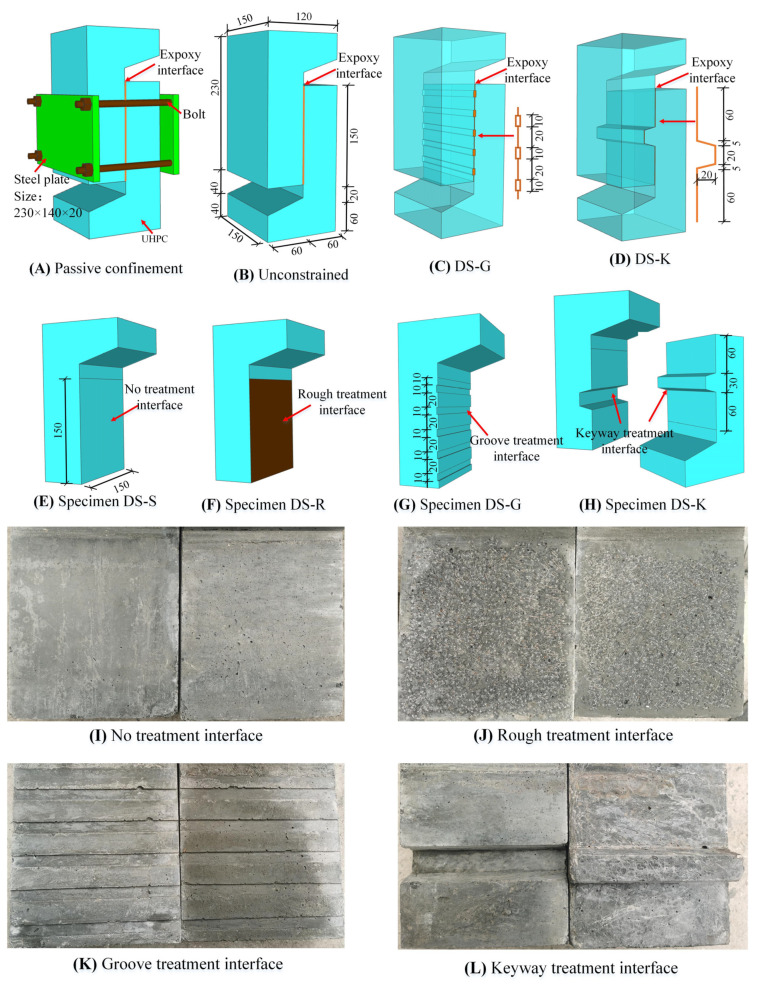
Details of the specimen: (**A**,**B**) schematic diagrams of passive restraint and unrestrained loading; (**C**,**G**) components and overall structure of the specimens DS-G and PC-DS-G; (**D**,**H**) the components and overall design of the test pieces DS-K and PC-DS-K; (**E**,**F**) parts of specimens DS-S, DS-R, PC-DS-S, and PC-DS-R; (**I**–**L**) the specimens after smooth, rough, groove, and key groove interface treatment, respectively. Unit: mm.

**Figure 2 polymers-14-01364-f002:**
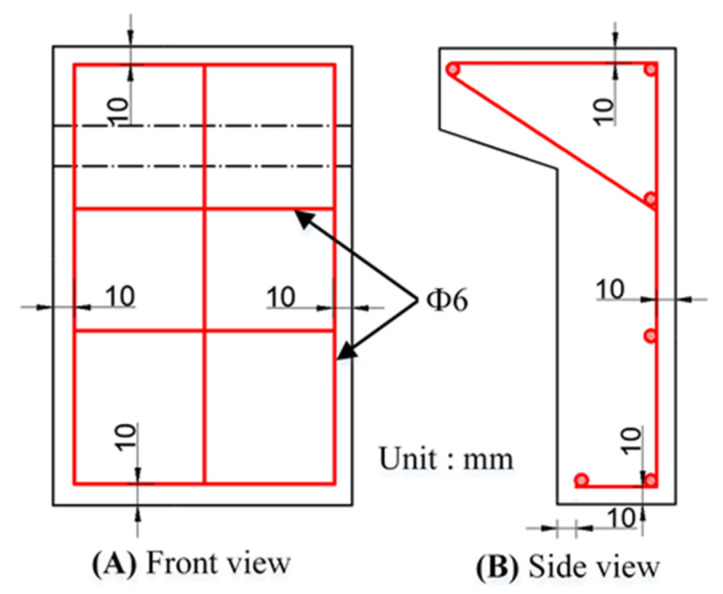
Reinforcing diagram of the specimen.

**Figure 3 polymers-14-01364-f003:**
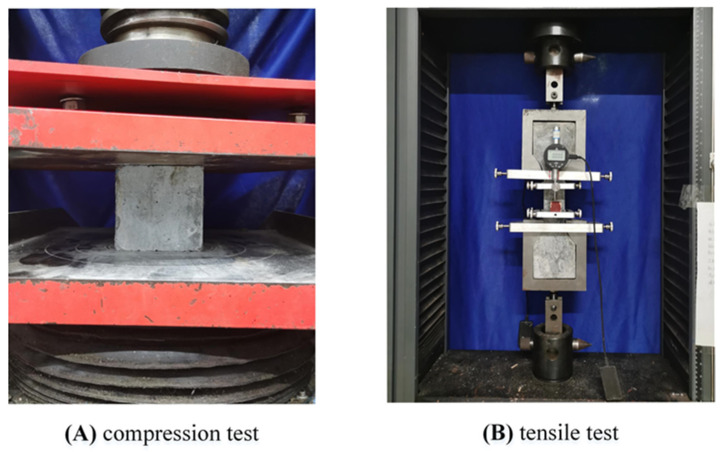
Material performance test of the UHPC.

**Figure 4 polymers-14-01364-f004:**
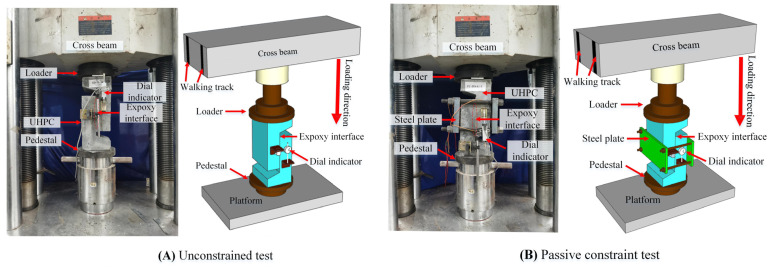
Test loading diagram.

**Figure 5 polymers-14-01364-f005:**
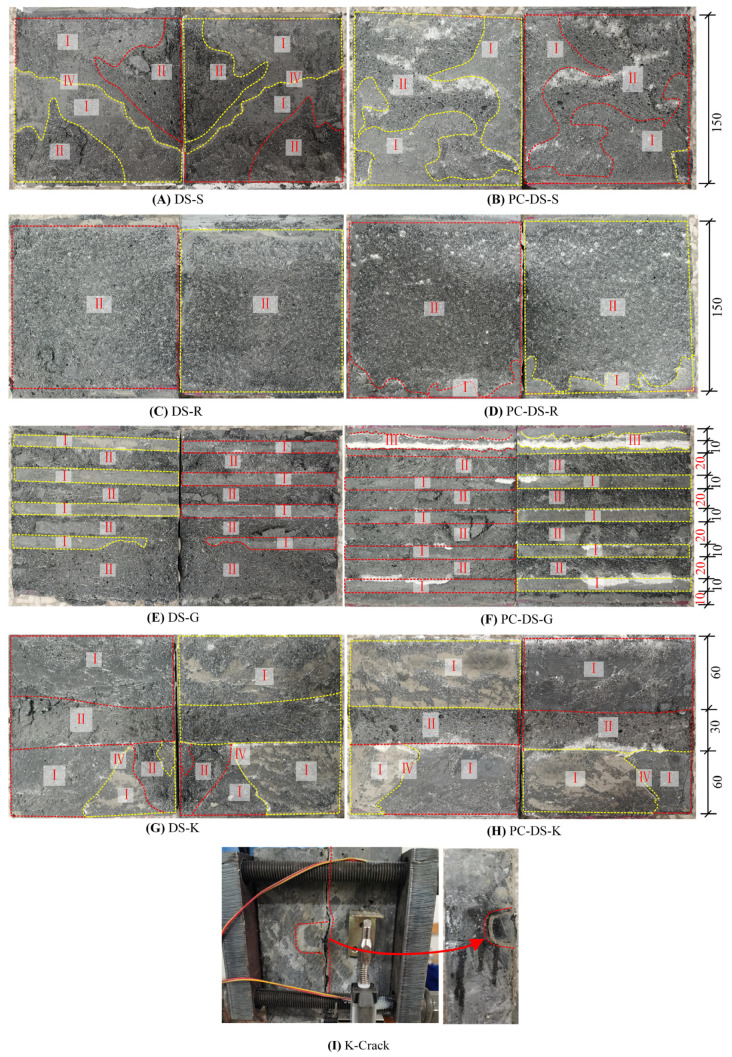
Interface damage state. Unit: mm.

**Figure 6 polymers-14-01364-f006:**
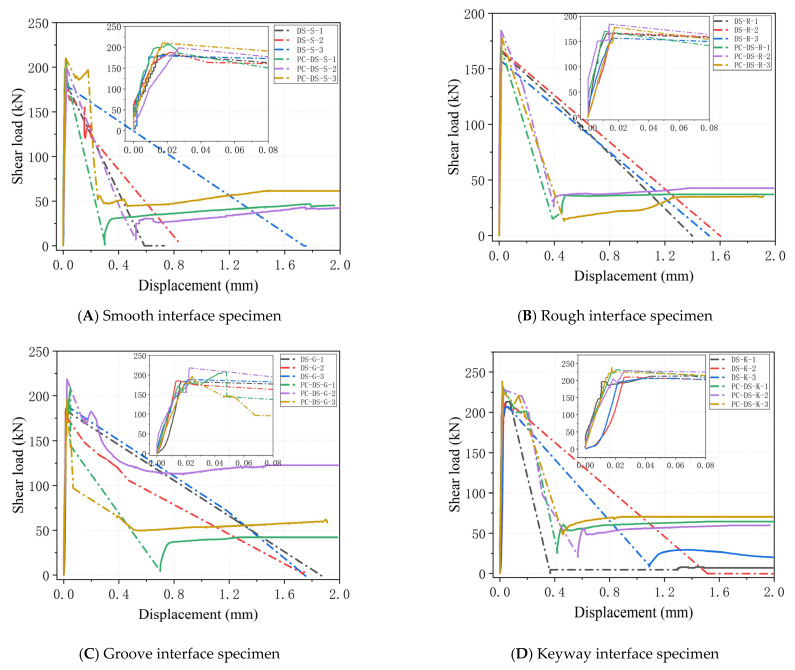
Load-displacement curves of different interfaces under unconstrained and passive constraints.

**Figure 7 polymers-14-01364-f007:**
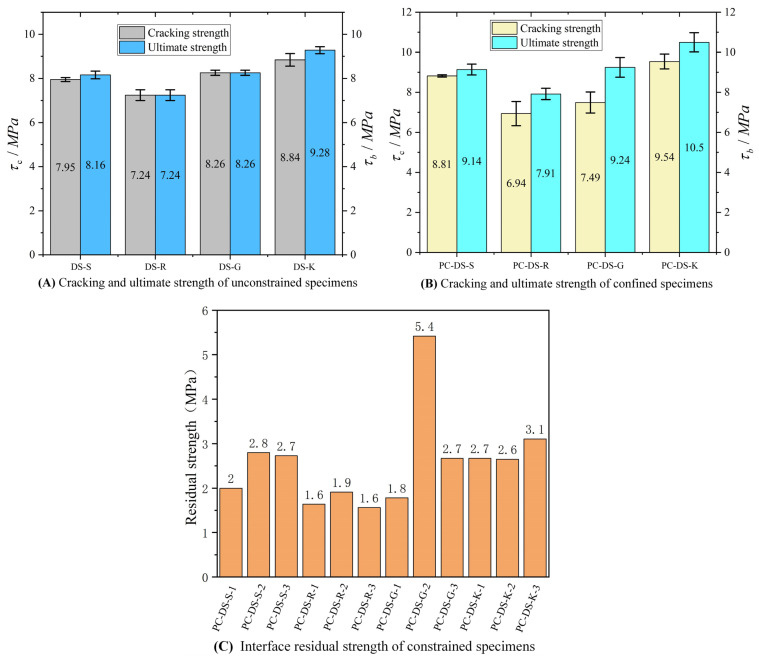
Interface Cracking, Limit, and Residual Strength.

**Figure 8 polymers-14-01364-f008:**
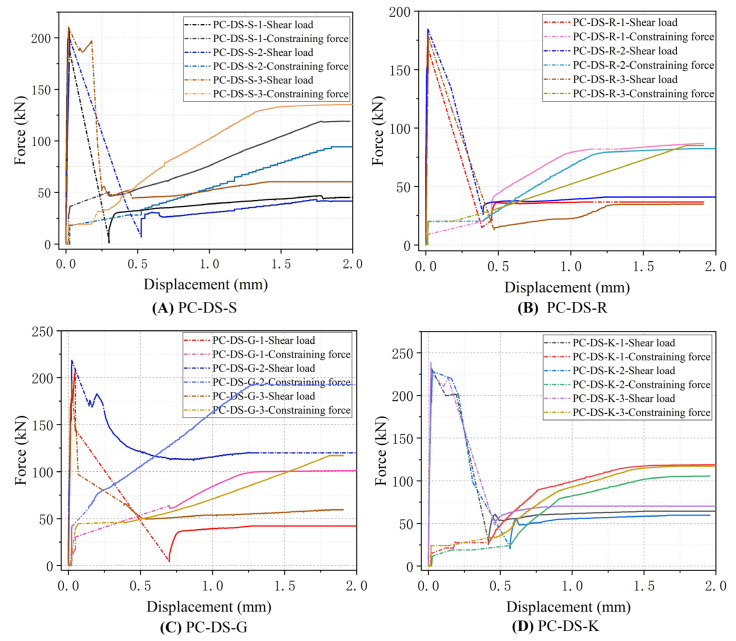
The change of load and passive constraint force with the increase of slip.

**Figure 9 polymers-14-01364-f009:**
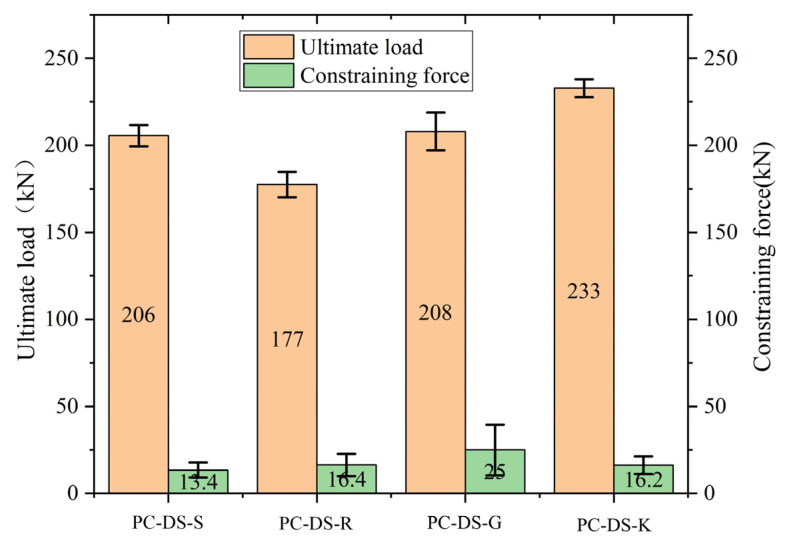
The ultimate bearing capacity and passive constraint force of the specimen.

**Figure 10 polymers-14-01364-f010:**
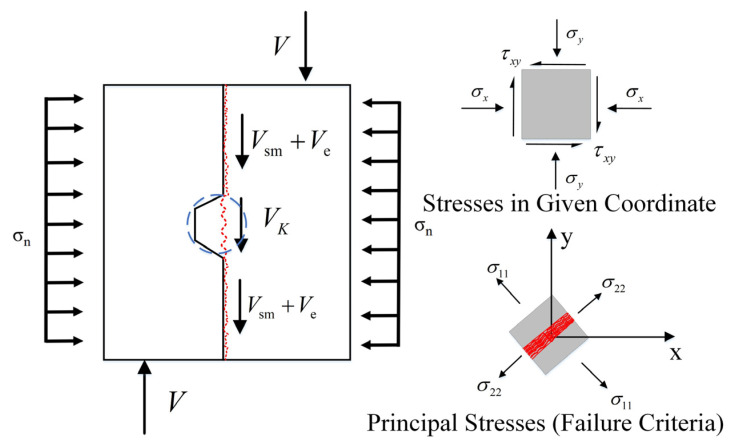
Proposed shear model for UHPC precast keyed joints.

**Figure 11 polymers-14-01364-f011:**
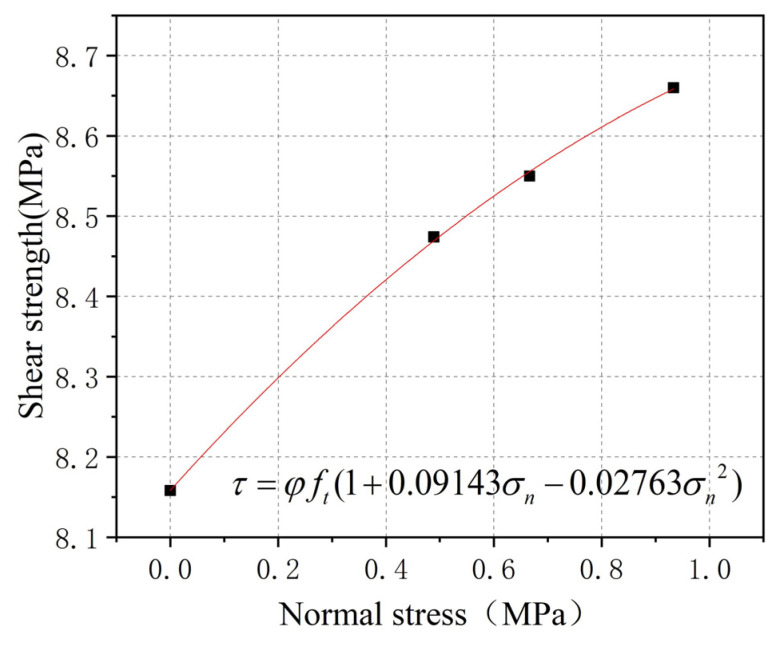
Fitting curve diagram.

**Figure 12 polymers-14-01364-f012:**
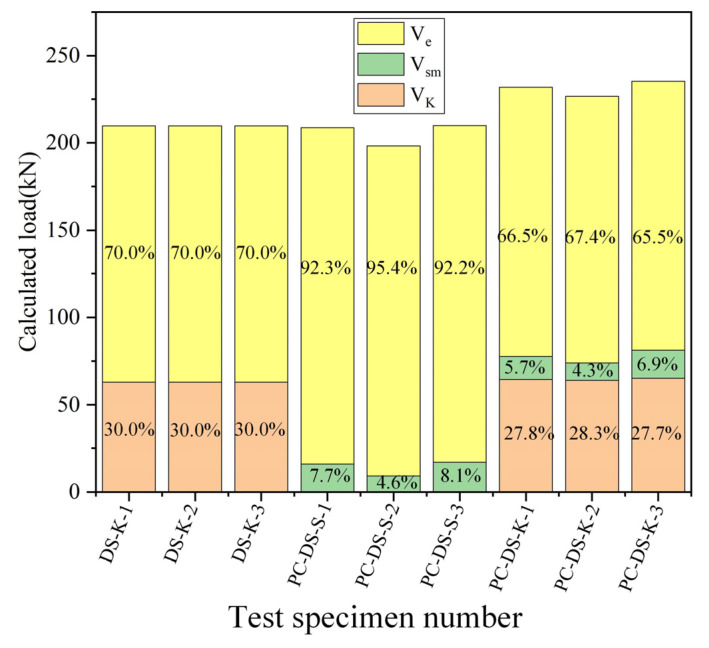
Contributions of each part of the interface.

**Table 1 polymers-14-01364-t001:** Specimen details.

Specimen Category	Interface Handling	Loading Method	Interface Size (mm)	Interface Epoxy Thickness (mm)	Number of Test Pieces
DS-S	No treatment	Unconstrained	150 × 150	2	3
DS-R	Rough treatment	Unconstrained	150 × 150	2	3
DS-G	Groove treatment	Unconstrained	150 × 150	2	3
DS-K	Keyway treatment	Unconstrained	150 × 150	2	3
PC-DS-S	No treatment	passive constraint	150 × 150	2	3
PC-DS-R	Rough treatment	passive constraint	150 × 150	2	3
PC-DS-G	Groove treatment	passive constraint	150 × 150	2	3
PC-DS-K	Keyway treatment	passive constraint	150 × 150	2	3

**Table 2 polymers-14-01364-t002:** Mix the proportion of UHPC.

Component	Mass Ratio	Proportion (%)
Premixed dry material	10.000	82.09
Steel fiber	1.2232	10.04
Water-reducing admixture	0.0672	0.552
Water	0.8916	7.318

**Table 3 polymers-14-01364-t003:** Mechanical properties of UHPC and epoxy resin.

Material	$f_{c}\;(\mathrm{MPa})$	$f_{ct}\;(\mathrm{MPa})$	$f_{t}\;(\mathrm{MPa})$	$E_{c}\;(\mathrm{MPa})$	$v_{c}$
UHPC	150	20	14.0	42100	0.2
CBSR-A/B	90	45	30	3200	/

Note: $f_{c}$ is the compressive strength; $f_{ct}$ is the flexural strength; $f_{t}$ is the tensile strength; $E_{c}$ is Young’s modulus; $v_{c}$ is the Poisson’s ratio of UHPC.

**Table 4 polymers-14-01364-t004:** Loading results of specimens.

Type	No.	$V_{c}\;(\mathrm{kN})$	$\delta_{c}\;(\mathrm{mm})$	$V_{b}\;(\mathrm{kN})$	$\delta_{b}\;(\mathrm{mm})$	$F_{p}\;(\mathrm{kN})$	$\tau_{c}\;(\mathrm{MPa})$	$\tau_{b}\;(\mathrm{MPa})$	$K_{c}\;(\mathrm{kN}/\mathrm{mm})$	$K_{b}\;(\mathrm{kN}/\mathrm{mm})$
DS-S	1	180.16	0.014	183.16	0.017	/	8.007	8.140	571.930	478.86
	2	176.54	0.017	187.70	0.022	/	7.846	8.342	461.54	379.19
	3	179.86	0.016	179.86	0.016	/	7.994	7.994	499.613	499.613
DS-R	1	165.54	0.012	165.54	0.012	/	7.357	7.357	603.163	603.064
	2	166.63	0.016	166.63	0.016	/	7.406	7.406	448.735	448.835
	3	156.68	0.016	156.68	0.016	/	6.964	6.964	424.607	424.607
DS-G	1	183.24	0.018	183.24	0.018	/	8.144	8.144	452.444	452.456
	2	185.61	0.015	185.61	0.015	/	8.249	8.249	549.943	549.943
	3	188.58	0.022	188.58	0.022	/	8.382	8.382	389.839	389.839
DS-K	1	196.67	0.011	213.56	0.075	/	8.741	9.491	794.626	116.8
	2	208.39	0.026	210.45	0.035	/	9.264	9.353	361.892	267.23
	3	193.89	0.021	206.87	0.056	/	8.617	9.194	404.571	154.750
PC-DS-S	1	199.40	0.014	207.71	0.021	15.4	8.862	9.231	656.460	435.429
	2	198.60	0.027	198.62	0.027	8.52	8.827	8.827	330.586	330.586
	3	197.01	0.015	210.38	0.017	16.3	8.756	9.350	572.267	537.384
PC-DS-R	1	169.95	0.011	169.95	0.011	9.0	7.553	7.553	686.667	686.667
	2	152.35	0.010	184.43	0.014	20.2	6.771	8.197	677.097	585.492
	3	146.08	0.014	178.04	0.017	19.9	6.492	7.913	463.746	452.157
PC-DS-G	1	172.19	0.015	208.95	0.048	17.8	7.653	9.287	524.170	193.069
	2	155.30	0.014	218.41	0.023	41.7	6.902	9.707	493.006	431.407
	3	178.06	0.019	196.58	0.024	15.5	7.914	8.737	420.943	358.067
PC-DS-K	1	216.87	0.019	231.75	0.203	15.8	9.64	10.30	521.009	508.641
	2	205.38	0.018	228.29	0.026	11.5	9.128	10.15	497.463	390.987
	3	221.59	0.016	238.42	0.018	21.4	9.848	10.59	618.224	591.131

Note: $V_{c}$ is the bearing capacity of various specimens when they are cracked, Unit: kN; $V_{b}$ is the ultimate bearing capacity of multiple specimens, Unit: kN; $\delta_{c}$ is the slippage of various specimens when they are cracked, Unit: mm; $\delta_{b}$ is the slippage of the specimen in the ultimate load state, Unit: mm; $F_{p}$ is the passive binding force when the load reaches $V_{b}$, Unit: kN; $\tau_{c}$ is the interface strength corresponding to the bearing capacity $V_{c}$ at cracking, Unit: MPa; $\tau_{b}$ is the interface strength corresponding to the ultimate bearing capacity $V_{b}$, Unit: MPa; $K_{c}$ is the interface secant stiffness corresponding to the interface cracking, Unit: kN/mm; $K_{b}$ is the interface secant stiffness corresponding to the interface limit state, Unit: kN/mm.

**Table 5 polymers-14-01364-t005:** Comparison of Formula and Experimental Results.

Part	Numbering	*V_k_*	*V_sm_*	*V_e_*	*V*	*V_d_*	*V*/*V_d_*
DS-K	1	63	/	146.8	209.8	213.562	0.982
	2	63	/	146.8	209.8	210.45	0.996
	3	63	/	146.8	209.8	206.876	1.014
PC-DS-S	1	/	16.03	192.66	208.7	207.7	1.005
	2	/	9.186	189.17	198.364	198.6	0.998
	3	/	16.89	193.04	209.4	210.386	0.995
PC-DS-K	1	64.56	13.131	154.2	231.9	231.75	1.001
	2	64.14	9.771	152.64	226.5	228.288	0.992
	3	65.1	16.16	154.1	235.3	238.418	0.986
Mean value							0.996
Standard deviation							0.027

Note: *V* represents the calculated value in the table, and *V_d_* represents the test value.

## Data Availability

The original data of the results of this study are archived. If necessary, the editorial department can contact yukun@mails.cqjtu.edu.cn by email.

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
