# Peer review of "Interfacial Shear Performance of Epoxy Adhesive Joints of Prefabricated Elements Made of Ultra-High-Performance Concrete"

_polymers, 2022, doi:10.3390/polym14071364_

Round 1
Reviewer 1 Report
Comments
This paper investigate the interfacial shear performance of epoxy adhesive joints. The outcome of the paper is interesting however, there are several aspects that need to be improved. The reviewer can onlyrecommend for publication if the author satisfactorily address the following major comments in the revised version.
- Suggest to add error bars in Fig 7.
- The research gap from the literature review should be clearly presented.
- The research questions and justification of selecting variables should be highlighted.
- Which test standards was considered in this study? How many replicate samples were tested in each category?
- The failure mechanism of the specimen should be discussed more clearly.
- The novelty of the study should be highlighted more clearly at the end of introduction section. How this study is different from the published study in literature?
- How the outcome of this study will benefit researchers and end users? This need to be highlighted in introduction or end of conclusion.
- The adhesive joint is interesting but not novel. Therefore, the recent application in this area should be discussed in introduction section to improve the background study. Recently, adhesive joint was considered for wall system [Ref: Axial compression behaviour of all-composite modular wall system] and sandwich panels [Ref: Bond behaviour of composite sandwich panel and epoxy polymer matrix: Taguchi design of experiments and theoretical predictions]. Suggest to include them in introduction section with proper citations to improve the background study.
I would be happy to see the revised version to understand how these comments are being addressed.
Author Response
All comments suggested by the reviewers have been accepted and revised in the manuscript.The changes made are as follows:
Point 1: Suggest to add error bars in Fig 7.
Response 1: The author adds an error bar to Figure 7.
Point 2:The research gap from the literature review should be clearly presented.
Response 2: The author has carried on the revision in the introduction part, has highlighted with the literature review research disparity.
Point 3:The research questions and justification of selecting variables should be highlighted.
Response 3: The the last part of the introduction and the beginning of the experiment, the author made modifications, emphasizing the research questions and the selection of variables.
Point 4:Which test standards was considered in this study? How many replicate samples were tested in each category?
Response 4: The author set up eight groups of experiments according to two loading methods and four interface types. Each group has three repeated specimens. For details, please see section 2.1 red modification section.
Point 5:The failure mechanism of the specimen should be discussed more clearly.
Response 5: The failure mechanism of the specimen was discussed more clearly. For details, please refer to the red section modified in Section 3.1 of the article.
Point 6:The novelty of the study should be highlighted more clearly at the end of introduction section. How this study is different from the published study in literature?
Response 6: The authors modified the end part of the introduction, highlighting the novelty of this study. For the time being, no scholars have considered the influence of passive constraints on interfacial shear properties in published studies.
Point 7:How the outcome of this study will benefit researchers and end users? This need to be highlighted in introduction or end of conclusion.
Response 7: The author has been modified at the end of the introduction. This study provides a calculation basis and reference for designers and researchers in the field of UHPC precast adhesive joints.
Point 8:The adhesive joint is interesting but not novel. Therefore, the recent application in this area should be discussed in introduction section to improve the background study. Recently, adhesive joint was considered for wall system [Ref: Axial compression behaviour of all-composite modular wall system] and sandwich panels [Ref: Bond behaviour of composite sandwich panel and epoxy polymer matrix: Taguchi design of experiments and theoretical predictions]. Suggest to include them in introduction section with proper citations to improve the background study.
Response 8: The author revised the introduction, and included two literatures in the introduction, which improved the research background.
Thank you very much for your valuable comments.
Best regards,
All authors
Reviewer 2 Report
The article is related to the interfacial shear performance of epoxy adhesive joints of prefabricated elements made of ultra-high performance concrete. The manuscript is well written. I have just soe minor remarks before further processing:
- I suggest to change the title to describe the phenomena of using ultra-high performance concrete in the research. The proposed title is: “Interfacial shear performance of epoxy adhesive joints of prefabricated elements made of ultra-high performance concrete”,
- please define abbreviation UHPC first time used in the abstract,
- please do not use citation pockets (e.g. [7-11] or [15-17]) but rather cite and discuss each reference individually (e.g. as mentioned in [8] etc.). This will allow reducing unnecessary references,
- I would like to see scale bars on fig. 1 and 5 to see the size of the elements and surfaces,
- I suggest to merge tables 3 and 4 into one table,
- I would like to see the error bars on fig. 7 and fig. 9 to see the scatter,
- I thin that it will be beneficial to add some perspectives in conclusion section.
Author Response
All comments suggested by the reviewers have been accepted and revised in the manuscript.The changes made are as follows:
Point 1: I suggest to change the title to describe the phenomena of using ultra-high performance concrete in the research. The proposed title is: “Interfacial shear performance of epoxy adhesive joints of prefabricated elements made of ultra-high performance concrete”, -æˆ‘å»ºè®®å°†æ ‡é¢˜æ”¹ä¸º
Response 1: The author has changed the title to“Interfacial shear performance of epoxy adhesive joints of prefabricated elements made of ultra-high performance concrete”.
Point 2: please define abbreviation UHPC first time used in the abstract,
Response 2: The authors define in the abstract the abbreviations first used by UHPC.
Point 3: please do not use citation pockets (e.g. [7-11] or [15-17]) but rather cite and discuss each reference individually (e.g. as mentioned in [8] etc.). This will allow reducing unnecessary references,
Response 3: The authors revised the citation bag and made individual citations based on the main content of the literature.
Point 4: I would like to see scale bars on fig. 1 and 5 to see the size of the elements and surfaces,
Response 4: The author has been modified in Figure 1 and Figure 5, and added the necessary dimensioning.
Point 5: I suggest to merge tables 3 and 4 into one table,
Response 5: The author has merged tables 3 and 4 into one table.
Point 6: I would like to see the error bars on fig. 7 and fig. 9 to see the scatter,
Response 6: The authors have added error bars in Fig. 7 and Fig. 9.
Point 7: I think that it will be beneficial to add some perspectives in conclusion section.
Response 7: The author has added some opinion conclusions to the conclusion. The author believes that the passive binding force will change with the loading, and finally the passive binding force and the bearing capacity after failure will tend to a relatively stable ratio.
Thank you very much for your valuable comments.
Best regards,
All authors
Reviewer 3 Report
Dear Authors,
The topic of the paper is interesting and suits the Journal of MDPI Polymers. However, a major revision is required before this manuscript is qualified to be published in this prestigious journal. The manuscript is needed to be revised grammatically. The authors are required to check the whole manuscript with a grammar specialist as it has several grammatical errors. Only after revising the manuscript based on the comments, the paper is suggested to be published in MDPI. Further information on various issues identified in the manuscript appears below:
- The introduction section needs to be revised. A paragraph should be dedicated to the importance of your work.
- Please provide more detailed reasoning behind the behavior. The details should include rigid numbers or percentages.
- Please indicate how many samples for each experiment have been used. Please revise the other experiments respectively.
- Please add error bars to the figures where possible.
- Please describe the process of each experiment. Also indicate the model of each tool that is used in the experiment. What is the accuracy of each machine? Please explain them accurately.
- The conclusion needs more elaboration. Please use more sentences containing percentages and illustrate the main conclusions in the manuscript. Please paraphrase your results and discussions and use them in the conclusion part.
Author Response
All comments suggested by the reviewers have been accepted and revised in the manuscript. The changes made are as follows:
Point 1: The introduction section needs to be revised. A paragraph should be dedicated to the importance of your work.
Response 1: The author added the importance of the study by modifying the penultimate paragraph of the introduction.
Point 2: Please provide more detailed reasoning behind the behavior. The details should include rigid numbers or percentages.
Response 2: The author has carried on the revision in section 3.1, has carried on the detailed mechanism analysis. A comparative analysis was conducted in section 3.3.
Point 3: Please indicate how many samples for each experiment have been used. Please revise the other experiments respectively.
Response 3: The experiments section has been modified by the authors, and the authors have set up eight sets of experiments with three replicates each based on two loading modes and four interface types.
Point 4: Please add error bars to the figures where possible.
Response 4: The authors add error bars in Fig. 7 and Fig. 9.
Point 5: Please describe the process of each experiment. Also indicate the model of each tool that is used in the experiment. What is the accuracy of each machine? Please explain them accurately.
Response 5: The author has modified the test part and explained the accuracy of the tools used in the test.
Point 6: The conclusion needs more elaboration. Please use more sentences containing percentages and illustrate the main conclusions in the manuscript. Please paraphrase your results and discussions and use them in the conclusion part.
Response 6: The author has modified the conclusion and added the viewpoint that the passive constraint force changes with the loading, and the contribution proportion of each part of the interface, etc. conclusions.
Thank you very much for your valuable comments.
Best regards,
All authors
Round 2
Reviewer 1 Report
I have no further comments